

# Advancing measurements and representations of subsurface heterogeneity and dynamic processes: towards 4D hydrogeology

Thomas Hermans[1], Pascal Goderniaux[2], Damien Jougnot[3], Jan Fleckenstein[4], Philip Brunner[5], Frédéric Nguyen[6], Niklas Linde[7], Johan Alexander Huisman[8], Olivier Bour[9], Jorge Lopez Alvis[6,a], Richard Hoffmann[6,8], Andrea Palacios[10,b], Anne-Karin Cooke[11,12,c], Álvaro Pardo-Álvarez[5], Lara Blazevic[3,d], Behzad Pouladi[9,e], Peleg Haruzi[8,13], Meruyert Kenshilikova[9], Philippe Davy[9], Tanguy Le Borgne[9]

[1] Department of Geology, Ghent University, 9000 Gent, Belgium

[2] Department of Geology and Applied Geology, University of Mons, 7000 Mons, Belgium

[3] UMR 7619 METIS, Sorbonne Université, UPMC Université Paris 06, CCNRS, EPHE, F-75005 Paris, France

[4] Department of Hydrogeology, Helmoltz Centre for Environmental Research, 04318 Leipzig, Germany

[5] Laboratory of Hydrogeological Processes, University of Neuchâtel, 2000 Neuchatel, Switzerland

[6] Urban and Environmental Engineering, Liege University, 4000 Liege, Belgium

[7] Institute of Earth Sciences, University of Lausanne,1015 Lausanne, Switzerland

[8] Agrosphere (IBG 3), Institute of Bio- and Geosciences, Forschungszentrum Jülich GmbH, 52425 Jülich, Germany

[9] Geosciences Rennes, UMR 6118, Université de Rennes 1, CNRS, 35000 Rennes, France.

[10] Institute of Environmental Assessment and Water Research (IDAEA), Consejo Superior de Investigaciones Científicas (CSIC), Barcelona, 08034, Spain

[11] Géosciences Montpellier, University of Montpellier, CNRS, Univ. des Antilles, 34095 Montpellier, France

[12] Institut d'Optique d'Aquitaine, Muquans, 33400 Talence, France

[13] Department of Environmental Physics and Irrigation, Agricultural Research Organization – Volcani Institute, 7505101 Rishon LeZion, Israel

[a] Now at Centro de Geociencas , Universidad Nacional Autonoma de Mexico, 76230 Querétaro, Mexico.

[b] Now at Amphos 21 Consulting, 08019 Barcelona, Spain.

[c] Now at Federal Institute for Geosciences and Natural Resources (BGR), 13593 Berlin, Germany

[d] Now at Ruden SA, 0349 Oslo, Norway

[e] Now at Silixa Ltd, London, UK.

*Correspondence to*: Thomas Hermans (Thomas.hermans@ugent.be)

**Abstract.** Essentially all hydrogeological processes are strongly influenced by the subsurface spatial heterogeneity and the temporal variation of environmental conditions, hydraulic properties, and solute concentrations. This spatial and temporal variability needs to be considered when studying hydrogeological processes in order to employ adequate mechanistic models



or perform upscaling. The scale at which a hydrogeological system should be characterized in terms of its spatial heterogeneity
and temporal dynamics depends on the studied process and it is not always necessary to consider the full complexity. In this
paper, we identify a series of hydrogeological processes for which an approach coupling the monitoring of spatial and temporal
variability, including 4D imaging, is often necessary: (1) groundwater fluxes that control (2) solute transport, mixing and
reaction processes, (3) vadose zone dynamics, and (4) surface-subsurface water interaction occurring at the interface between
different subsurface compartments. We first identify the main challenges related to the coupling of spatial and temporal
fluctuations for these processes. Then, we highlight some recent innovations that have led to significant breakthroughs in this
domain. We finally discuss how spatial and temporal fluctuations affect our ability to accurately model them and predict their
behavior. We thus advocate a more systematic characterization of the dynamic nature of subsurface processes, and the
harmonization of open databases to store hydrogeological data sets in their four-dimensional components, for answering
emerging scientific question and addressing key societal issues.

**Short summary.** Although invisible, groundwater plays an essential role for the society as a source of drinking water or for
ecosystems by providing baseflow to rivers, but is also facing important challenges in term of contaminations. Characterizing
groundwater reservoirs with their spatial heterogeneity and their temporal evolution is therefore crucial for their sustainable
management. In this paper, we review some important challenges and recent innovations in imaging and modelling
groundwater reservoirs.

## 1 Introduction

While the surface components of continental water, such as streams, lakes and glaciers, are a very familiar part of our
landscape, the vast majority of continental water resources resides and flows in the subsurface, and is thus generally
inaccessible to direct observation (McDonnell, 2017). Growing societal needs imply that subsurface environments, which form
part of the critical zone of the Earth (Brantley et al., 2007; Fan et al., 2019), are increasingly subject to pressure and multiple
(possibly competing) uses for water resources such as groundwater abstraction, artificial recharge and storage (Dillon et al.,
2019; Russo and Lall, 2017; Aeschbach-Hertig and Gleeson, 2012), nuclear waste storage (e.g., Ewing, 2015, Butler, 2010),
geothermal energy (Rivera et al., 2017; Fleuchaus et al., 2018; Lu, 2018), oil and gas extraction (e.g., Wang et al., 2014) and
climate change mitigation such as energy storage (Arbabzadeh et al., 2019) and $CO_2$ sequestration (Hamza et al., 2021, Kumar
et al., 2020) while being threatened by anthropogenic contamination (e.g., Riedel et al., 2020). As a result, subsurface systems
are experiencing profound modifications that affect their basic environmental functions and ecosystem services (Erostate et
al., 2020; Fattorini et al., 2020; Luijendijk et al., 2020). These modifications include, both at the local and the catchment scales,
water level depletion (Jasechko et al., 2021), which affects baseflow of many rivers and associated ecosystem services (Conant
et al., 2019), a growing input of chemicals and pathogens, which threaten water quality (e.g., Szymczycha et al., 2020),
seawater intrusion (Werner et al., 2013) and soil salinization (Litalien and Zeb, 2020; Singh, 2021) threatening soil- and water



resources as well as food security in many arid and semi-arid regions of the world, and massive fluid injections at depth related to $CO_2$ sequestration or gas extraction, which may lead to increased seismicity (Rathnaweera et al., 2020; Schultz et al., 2020; Keranen and Weingarten, 2018).

The last decade has seen great advances in stochastic subsurface hydrology (e.g., Scheidt et al., 2018), microscale imaging
(e.g., Gouze et al. 2008, Blunt et al. 2013, Heyman et al. 2020), and geophysical characterization (e.g., Binley at al., 2015; Singha et al., 2015). Although geological heterogeneity has been long recognized, these advances have made it even more clear that the subsurface is highly heterogeneous at multiple scales and that this heterogeneity substantially controls many flow, transport and biochemical processes (e.g., Hartmann et al., 2017; Comte et al., 2019 Zamrsky et al., 2020). Recent efforts have led to an improved ability for monitoring surface-water dynamics or characterizing the state of aquatic systems, but this
has not been matched by a significant increase in our ability to quantify the dynamics of fluxes and processes in the subsurface (e.g., Schilling et al., 2018). A wide gap between common modelling approaches (e.g., homogeneous or multi-Gaussian representations of parameters, steady state or transient simulations, upscaling approaches) and field reality prevails. On the one hand, data sets often have a very limited 3D spatial extent and are characterized by a low sampling density, preventing a full description of the complex nature of the aquifer (e.g., Xu and Valocchi, 2015). On the other hand, studies concerning the
temporal dynamics of hydrological processes and structures are generally based on point data typically acquired in wells in hydrogeology, potentially missing the underlying spatial variability (e.g., Johnson et al., 2012). The persistent observation gap between data points results in a lack of knowledge and contributes significantly to the current lack of understanding of subsurface processes and our ability to accurately predict the evolution of subsurface systems.

This deficiency in the characterization of aquifer limits our ability to answer critical scientific questions of significant societal
and industrial impacts. For example, the magnitude, spatial distribution and temporal dynamics of fluxes between subsurface and surface compartments of the Earth are relevant for the management of water quantity, quality and ecology (Fleckenstein et al., 2006, 2010; Brunner et al., 2017; Conant et al., 2019). Similarly, the dispersion and residence time distribution of dissolved chemical species in heterogeneous porous and fractured media, as well as the coupling between fluid mixing and reactions and the localization of hotspots of subsurface biogeochemical reactions, are key aspects to study the fate of
contaminations (e.g., Wallis et al., 2020; Pannecoucke et al., 2020; Robinson and Hasenmueller, 2017; Bailey, 2017) or understand the contribution of subsurface processes to global cycles of carbon (Zhang and Planavsky, 2020; Liu et al., 2014) and nitrogen (Marzadi et al., 2012) (Figure 1). The above-mentioned processes are both highly dynamical and strongly influenced by small-scale and across-scale heterogeneity (e.g., Salehikhoo et al., 2013, Dentz et al., 2011). In particular, the location and reactivity of biogeochemical hot spots (e.g. McClain et al. 2003), which are thought to have a disproportionate
influence on macroscale critical zone processes relative to their size, may be largely influence by spatial heterogeneity coupled to temporal fluctuations (e.g. Rolle and Le Borgne 2019). The potential of a characterization for coupled spatio-temporal monitoring of parameters and state variables and their temporal evolution, including 4D imaging, to understand these processes remains largely unexplored. This temporal component should not only include the evolution of state variables under transient conditions, but also the evolution of system properties because of coupled processes such as hydromechanical effects impacting





the pore space or fracture apertures (Davy et al., 2018), colmatation and erosion processes in streambeds (Partington et al., 2017) or reactive transport inducing changes in the pore space (Izumoto et al., 2020).

A grand challenge of subsurface imaging methods for dynamic hydrogeological processes (Figure 1) is to deal with systems characterized by pronounced structure and process heterogeneity, including preferential flow paths, evolving properties or geometry, unsaturated flow processes, fluctuating redox conditions, and multifunctional microbial communities. Recent breakthroughs in hydrogeophysical imaging techniques (Binley et al., 2015; Singha et al., 2015) and the emergence of interdisciplinary approaches combining new sensors such as fiber optics (Bense et al., 2016; Zhan, 2020), new experimental methodologies like ambient seismic noise correlation (Garambois et al., 2019) and coupled modelling techniques (e.g., Hinnell et al., 2010; Jardani et al., 2013; Linde and Doetsch, 2016) may profoundly change our vision and representation of the dynamics of processes that take place in these environments (Binley et al., 2015; St. Clair et al., 2015). However, monitoring and characterizing dynamical fluxes, transport, reactions and hydromechanical processes that evolve spatially in 3D with geophysical imaging is still in its infancy in environmental sciences and engineering. The efficiency of those new methods and the full complexity that emerges from their coupling can only be revealed through in-situ exploration with interdisciplinary approaches. In that sense, field observatories and case studies constitute a key component of the current research effort (e.g., Folch et al., 2020, Palacios et al., 2020).

Although heterogeneity influences all the processes occurring in the subsurface, an exhaustive characterization of the subsurface is not always necessary and strongly depends on the objective of the studies. Large-scale water balance approaches, average advection velocity or aquifer recharge through the vadose zone might be approximated with scarce hydrogeological data. However, the integration of recent advances in imagining the spatial and temporal variability in hydrological model is still in its early stage and has a huge potential to improve the mechanistic understanding of hydrological processes, the upscaling of models and the assessment of model simplifications. The objective of this opinion paper is, therefore, to identify and discuss when, why, and for which processes and applications the characterization of dynamic hydrogeological processes is crucial. We discuss the potential and value of exploring and monitoring dynamic in hydrogeology, including 4D imaging, for understanding groundwater fluxes, transport, mixing and reactions processes, soil moisture dynamics in the vadose zone and surface-subsurface water interactions (Figure 1). Based on a non-exhaustive overview of recent advances, we identify key scientific challenges and their relation to the heterogeneous and dynamic nature of the subsurface for each of the above-mentioned processes. We then highlight some recent breakthroughs which allowed to advance our understanding of these processes. We also discuss the feasibility, advances and challenges of numerical modelling of the identified processes in terms of 4D complexity as well as the important role of instrumented field observatories and corresponding case studies to tackle the scientific challenges and evaluate the performance and scope of recent innovations.



## 2 Key processes in hydrogeology and their 4D nature

In this section, we highlight hydrogeological processes at different scales for which the considerations of coupled spatial heterogeneity and temporal dynamics are important, present the main challenges related to the representation and inference of spatial and temporal variability and point towards a few recent innovations that could help to address these challenges in the future.

### 2.1 Groundwater fluxes

Inferring and modeling groundwater Darcy fluxes and fluid velocities is crucial in most hydrogeological processes and related applications, both for quantitative (e.g., water storage, groundwater discharge, residence time) and qualitative (e.g., contaminant transport, reaction and mixing processes) purposes. Pore-scale advection flow, along with the other transport processes and the influence of the macro-scale geological heterogeneity, controls propagation and spreading of natural or contaminant solutes, from fast transfers to late time tailings (e.g. Dentz et al. 2011; Hoffmann et al., 2020; Kang et al., 2015). In the context of risk assessment, measuring natural solutes or contaminant concentrations may be of limited value if not supported by quantitative flux rates allowing to estimate solute mass transfers (Brouyère et al., 2008). Groundwater fluxes drive mixing processes prevailing at aquifer interfaces such as subsurface-surface interactions in hyporheic zones or transition zones along coastal saltwater intrusion (e.g. Werner et al. 2013, Hester et al. 2017). They also influence hydrogeochemical and biogeochemical reactions by transporting reactants, such as nutrients, to these interfaces. They impact the feasibility of storage applications including the injection and recovery of heat or $CO_2$ in the subsurface (Niemi et al. 2017, Fleuchaus et al. 2018) and control residence times distributions across watersheds (Goderniaux et al., 2013).

The range of variation of expected groundwater fluxes may be very large, making them difficult to image, with their spatial and temporal variation. For decades, the basic approach consisted in first measuring the hydraulic conductivity and then predict fluxes, based on the hydraulic gradient. This approach however presents the serious disadvantage to average local variations and offers a limited understanding of the groundwater flux spatial distribution in heterogeneous media (Palmer, 1993; Brouyère et al., 2008; Jamin et al., 2015). In addition, many subsurface processes are time-dependent and exhibit an inherent periodicity. If deep systems are not expected to vary rapidly, shallow aquifers can exhibit fluctuations ranging from lower (e.g. multi-annual or seasonal recharge fluctuations) to higher frequencies (e.g. tide dependent saltwater intrusion, aquifer exploitation or artificial storage applications). The accurate quantification of groundwater flux rates including their spatial distribution and transient conditions is, therefore, needed to understand and manage these processes. This should be performed at the relevant scale(s), with the appropriate resolution, in adequation with the objectives of the study and the geological context (Jimenez-Martinez et al., 2013).

Recent research efforts have focused on the development of direct or indirect methods allowing for a more accurate assessment of groundwater fluxes, including their amplitude, spatial distribution and temporal dynamics, with application in highly heterogeneous media. As groundwater fluxes influence many processes, some approaches, discussed in the next sections, have





been developed for specific contexts, such as surface-subsurface interactions or solute transport, allowing indirect quantification of the fluxes, for example through tracer experiments (Popp et al., 2021). Some other techniques are more general and available to directly and accurately measure local groundwater fluxes (e.g., Jamin et al., 2015; Le Borgne et al.,

2006 ; Brouyère et al., 2008 ; Burnett et al., 2006). For instance, point measurements of darcy fluxes are classically done from dilution methods (Drost et al., 1968; Klotz et al., 1980; Pitrak et al., 2007; Novakowski et al., 2006; Jamin et al., 2015). Well-points velocity probes have been also recently developed (Labaky et al., 2009; Devlin, 2020) to provide promising and complementary velocity measurements although ranges of measurements are still limited. A key aspect of some recent approaches is to allow the monitoring of groundwater fluxes dynamics (Jamin and Brouyère, 2018). Important challenges, also

depending on the media type, however, remain. The number of available methods for direct measurement is limited and most methods are suited for porous media rather than fractured aquifers where fluxes are expected to show stronger spatial variations. Current methods are still deficient in providing full 3D high-resolution datasets as they must be performed in wells. Current geophysical imaging techniques are still unable to directly estimate fluxes, as they are based on the contrast in water properties induced through tracers. Nevertheless, very promising results were obtained through the combination of ambient

noise surface wave tomography (ANSWT), and self-potential (SP) measurements to image the complex hydrogeology structure and associated groundwater flow paths at a coastal site (Grobbe et al., 2021). The joint interpretation of the SP and seismic data permits to show that groundwater flow occurs in the identified paleo-channels at the erosional surface of the basaltic bedrock (Grobbe et al., 2021). Despite these improvements, monitoring the dynamics of fluxes remains resource expensive. Accurate measurements, which often imply complex experimental needs and designs, must be repeated regularly,

with an adequate time resolution, for example to understanding cyclic process evolution. Passive flux methods (e.g., Hatfield et al., 2004) integrate flux measurements over specific periods, providing mean representative value while avoiding repetitive field operations, but do not capture the dynamics.

Fiber Optic Distributed Temperature Sensing (DTS) allows for the measurement of temperature with high spatial and temporal resolution over large distances from a few hundreds m up to several km, using buried or borehole cables (Selker et al., 2006;

Bense et al., 2016; Simon et al., 2020, see table 1.C). By estimating the rates of temperature change along the cable, this kind of system allows an indirect estimation of groundwater fluxes intercepting the cable, provided that conditions are changing fast enough and that the temperature change is large enough to be detected. Read et al. (2013) for example resolved groundwater fluxes in fractured granite, by combining DTS measurements and hot water injections in a borehole. Long-term changes can also be detected by using DTS systems as permanent monitoring tools (Susanto et al., 2017; McCobb et al., 2018).

A new generation of active fiber-optics with heated cables designed for hydrogeological investigations is currently being developed and is a promising approach for inferring borehole or in-situ groundwater fluxes (Read et al., 2014; des Tombes et al., 2019; Maldaner et al., 2019; Simon et al., 2021; Del Val et al., 2021). The thermal response during the active heating of the cable and the subsequent cooling period is monitored as it depends strongly on the water fluxes intercepting the cable, allowing an accurate groundwater flux assessment (Simon et al., 2021). Although challenges still remain to deploy such set-





ups on the field, possible DTS applications extend to various domains, including 3D hydraulic tomography (Pouladi et al., 2021) or groundwater-surface water interactions (see section 2.4).

In summary, the spatial distribution and dynamics of fluxes constitute a key component to many hydrogeological processes but are still poorly resolved. Although integrating 4D aspects in flux assessment might not be needed in larger scale water balance studies, they still influence residence time at the catchment scale. The development of innovative in-situ methods to

characterize fluxes is an important element, unravelling the 4D distributions of fluxes remain a major challenge in hydrogeology for process understanding or environment characterization.

## 2.2 Transport, mixing and reaction

Three-dimensional heterogeneity and temporal fluctuations of fluxes have a first-order impact on transport and reaction processes (e.g. Dentz et al. 2011, Rolle and Le Borgne 2019, Valocchi et al. 2019). The inherent heterogeneity of subsurface

environments leads to strong dispersion, which do not follow the conventional Fickian dispersion framework (e.g. Berkowitz et al. 2006, Neuman and Tartakovsky 2009). Furthermore, mixing dynamics can be fundamentally different than predicted from 2D or steady representations of the subsurface (Lester et al. 2013). At the pore scale, recent 3D imaging techniques (Figure 2) have shown that 3D flow topologies driven by pore scale flow patterns lead to chaotic flows that strongly enhance mixing rates (Heyman et al. 2020, Souzy et al. 2020). At the Darcy scale, anistropic permeability fields can generate helical

flow that play a similar role (Ye et al. 2015). In fractured media, intersection of fractures with fluids of different chemical compositions can create microbial hot spots with intermittent activity (Bochet et al. 2020). Modelling and laboratory investigations have shown that these transport and reaction rates can be further altered by temporal fluctuations in head levels (Pool and Dentz, 2018) and variable water content (Jiménez-Martínez et al., 2017). Modelling studies have provided evidence that heterogeneity and temporal fluctuations can exert a strong control on biogeochemical reaction rates (Li et al., 2010; Sanz-

Prat et al., 2016). However, there is increasing evidence that reactive transport processes are not well captured by the macrodispersion framework (Gramling et al. 2002, de Anna et al. 2014). Yet, in the absence of an alternative upscaling framework, the macrodispersion model is still the main reference for field applications.

Characterizing and imaging transport and reaction dynamics in the field is a critical challenge for a range of fundamental and applied questions, such as designing efficient remediation strategies for contaminated sites (Kitanidis and McCarty, 2012) or

characterizing transport and reaction dynamics in mixing zones (Rolle and Le Borgne, 2019). Such reactive hot spots tend to develop at the interfaces between surface and subsurface compartments (Mcclain et al. 2003), which includes the vadose zone (Jimenez-Martinez et al., 2017, see section 2.3), the hyporheic zone (Hester et al., 2017, see section 2.4) or the groundwater-seawater interface (Pool and Dentz, 2018).

Classical artificial tracer tests are commonly used to estimate solute transport properties and related parameters. Their use in

highly heterogeneous media is, however, challenging due to the difficulty of positioning a limited amount of recovering points leading to low mass recovery (Kemna et al., 2002; Sanford et al., 2006). When interpreting or inverting the breakthrough data, with little information on the spatial heterogeneity, the range of possible interpretation in terms of parameter values can be





quite large or misleading (Hoffmann et al., 2019). Combining tracer test recovery with other monitoring methods, such as geophysics, and model inversion provides complementary information (e.g.,. Robert et al., 2012) that can narrow down the

uncertainty in the interpretation. Combining multiple tracers can also provide more constraints on tracer test interpretation (Hoffmann et al., 2019, 2021a,b, table 1.E). Tirado-Conde et al. (2019) combined 180 seepage meter measurements with heat used as a tracer in 30 locations, to study and characterize surface water-groundwater interactions and saltwater intrusion around a coastal lagoon inlet. Hoffmann et al. (2019), Klepikova et al. (2016) combined classical tracers with heat injection, taking advantage of heat conduction processes in rocks and sediments to better image the heterogeneity (Figure 3). They coupled

these different tracer experiments with electrical imaging (ERT) and fiber optics (Hermans et al., 2015b, table 1.A.1) to image highly heterogeneous alluvial deposits with high resolution (3), in the context of experimental pumping operations performed at short time-scale. DTS fiber optics techniques provide the opportunity for spatial monitoring of thermal tracers (de La Bernardie et al., 2018; Klepikova et al., 2016, see also section 2.1 and table 1.C.3). The combination of tracer experiments under different configurations (convergent, push pull,…) provides new constraints on transport models, highlighting the

possibility to capture the effect of complex 3D fracture networks architectures in effective transport models (Kang et al., 2015; Guilhéneuf et al., 2017).

New mobile mass spectrometers have opened up new opportunities to use dissolved gas as tracers and measure them continuously in the field (Brennwald et al., 2016; Chatton et al., 2017;Popp et al., 2021). Dissolved gases, such as Helium or Argon, are conservative tracers with larger diffusivity compared to solute tracers, thus, allowing the exploration of diffusive

processes such as fracture-matrix or mobile-immobile water interactions. Hoffmann et al. (2020) combined dye tracers with dissolved gases (Helium, Argon, Xenon) to study preferential flowpaths and mobile-immobile water effects within a dual porous/fractured rock chalk aquifer (table 1.E.2). Reactive tracers have offered new methods for characterizing transport dynamics, including hyporheic exchange (Knapp et al., 2017, see also section 2.4).

The use of time-lapse geophysical techniques provides a promising avenue to characterize the spatial distribution and temporal

evolution of transport and reaction processes, at scale up to a few hundred meters (e.g. Binley et al. 2015, table 1). Extensive geophysical imaging of transport processes has mainly been performed using Electrical Resistivity Tomography (ERT) and Ground Penetrating Radar (GPR) even if immediate successes have often been hampered by issues of mass recovery due to unresolved concentration gradients (Slater et al., 2002; Singha and Gorelick, 2005; Müller et al., 2010; Doetsch et al., 2012; Dorn et al., 2012b, Fernandez-Visentini et al. 2020). A major challenge is, thus, to upscale the non-stationary and non-ergodic

solute concentration fields as well as the macroscopic heterogeneity unresolved by geophysics (Gueting et al., 2015, 2017, table 1.D.2) to derive relevant petrophysical relationships. Accounting for realistic heterogeneity patterns in inversion remains difficult (both flow and transport) and upscaling is not straightforward (Singha et al., 2015). In smoothness-constrained tomography inversion, there is usually underprediction of magnitudes and overprediction of target sizes (Day-Lewis et al., 2006). Overcoming this challenge requires advanced hydrogeophysical imaging (Hermans et al., 2016b, 2018, Oware et al;,

2019), adapted regularization consistent with the studied process (e.g., Karaoulis et al., 2014; Hermans et al., 2016a, Nguyen et al., 2016; Lopez-Alvis et al., 2021), geostatistical post-processing (Moysey et al., 2005; Nussbaumer et al. 2019) or coupled





inversion (e.g., Hinnel et al. 2010). Recent modelling results suggest that key geostatisical properties of permeability fields may be inferred from time-lapse ERT imaging (Fernandez Visentini et al., 2020). New geophysically sensitive tracers, allowing density matching with the resident fluid (Shakas et al., 2017, table 1.B.3), provide images of tracer pathways that are not influenced by density effects, a key limitation of geophysical techniques based on dense saline tracers. Theoretical work has suggested that tracers that change their electrical conductivity when reacting could be imaged by electrical methods, providing new opportunities to characterize mixing processes in situ (Ghosh et al., 2018). This idea remains to be tested in the laboratory and field. Geophysical techniques that have the potential to map and monitor reactive processes, such as Spectral Induced Polarization (SIP), which consists in measuring the phase shift of an alternating electrical signal occurring because of polarization phenomena in the electrical double layer, by mineral precipitation (Leroy et al., 2017), or by the activity of microorganisms (Kessouri et al., 2019) are highly sensitive to pore scale processes and concentration distributions (Izumoto et al., 2020), which make their interpretation challenging but potentially very rewarding. The coupling of geophysical techniques with pore scale imaging techniques, including micro and millifluidics, represents a new avenue of research to understand and quantify the geophysical signature of unresolved pore scale processes (Jougnot et al., 2018; Fernandez Visentini et al., 2021).

Geophysical methods are also increasingly used for mapping biogeochemical processes (Atekwana and Slater, 2009; Knight et al., 2010). Self-potential (SP) signals have, under specific conditions, been shown to be sensitive to redox conditions in contaminated groundwater (Naudet et al., 2003; Revil et al., 2009; Arora et al., 2007). Laboratory studies have shown the correlation between SIP and bacteria activity using column experiments (Davis et al., 2006; Abdel Aal et al., 2010; Zhang et al., 2014) and the SIP method has been applied to detect biogeochemical reactions or root activities in the field (Wainwright et al., 2016; Flores Orozco et al., 2012; Ehosioke et al, 2020). However, current interpretations are largely qualitative or empirical through correlation. It remains challenging to mechanistically relate the SIP signal to biological and physiological processes or simply to the biomass itself. For many applications, the underlying mechanisms of the observed polarization are still subject of active research and debate (see e.g., Leroy et al. 2017, Ehosioke et al., 2020), while field applications remain limited (Flores Orozco et al., 2021).

The coupling of heterogeneity, transport and reaction often lead to scale effects influencing effective reactive transport parameters (Salehikhoo et al., 2013, Dentz et al., 2011), making upscaling a major challenge in transport characterization and reactive transport modelling at the catchment scale (Li et al., 2017). As discussed above, hydrogeophysical imaging of transport and reaction processes is very attractive, but it requires upscaling the effects of sub-scale transport dynamics to the scales resolved by geophysical techniques (Fernandez Vissentini et al., 2020). Processes occurring at unresolved scales require imaging by combining multiple methods across scales, different tracers, and use of integrating data with geostatistics, modeling and inversion (Linde and Doetsch, 2016). Similar challenges occur when imaging the water content distribution in the vadose zone as discussed in section 2.3.

Recent innovations have demonstrated promising research avenues for overcoming the above-mentioned challenges. Recent modelling and field work have provided a new understanding of the role of heterogeneity and temporal fluctuations in the



development of mixing and reaction hot spots and hot moments. In fractured rocks, three-dimensional fracture patterns trigger localized mixing of fluids of different chemical compositions, leading to highly reactive hot spots at fracture intersections that develop intermittently depending on hydrological fluctuations (Bochet et al., 2020). In coastal aquifers, mixing between freshwater and saline water trigger reactions, including rock dissolution that leads to increased permeability and karst

formation, which develop as hot spots due to medium heterogeneity (De Vriendt et al., 2020). Improved time and space resolution of geophysical techniques, inverse modeling advances to monitor time-dependent processes, and the development of systems capable of surveying large areas repeatedly with multiple hydrogeophysical methods open new perspectives for mapping and monitoring these dynamics in the field (Folch et al., 2020; Palacios et al., 2020, table 1.B.6).

In summary, there is increasing evidence that conventional sampling techniques are not able to capture mechanisms underlying

the 3D heterogeneity and temporal fluctuations playing a key role in transport and reaction processes. Recent advances combined with 4D imaging open new opportunities for the exploration of transport and reaction processes from lab to field scales, which will help to enable the development of new mechanistic models that effectively capture these dynamics and allow for accurate upscaling approaches.

## 2.3 Water content dynamics in the vadose zone

The vadose or unsaturated zone is the upper part of the critical zone. The distribution of fluid phases and their evolution with time makes it a very complex media, where root systems and soil micro-organisms further complexify this dynamic environment. Understanding, monitoring and predicting the quantity (i.e. water content) and movement (i.e., water flow) of water are needed to address water quality and availability issues (e.g., Vereecken et al., 2015). Vadose zone hydrology usually relies on punctual measurements of physical variables with established sensors: TDR (Time-Domain Reflectometry, to infer

the water content from dielectric permittivity) or tensiometers (to determine the matric potential). Vadose zone hydrology is still too often viewed by hydrogeologists as a vertical 1D transit compartment with homogeneous sources (rainwater infiltration) and sinks (evaporation or evapotranspiration) on the way to the aquifer. If such approach might be sufficient to estimate average aquifer recharge rate, punctual measurements and 1D modelling approaches provide limited information for the characterization of this strongly 3D environment and its dynamics. The varying water content (in time and space) and

biological interactions (e.g. with the roots) are adding a layer of complexity compared to the saturated zone, making its spatial and dynamical characterization even more challenging.

While geophysical methods can provide fast and integrated measurements to characterize the spatial heterogeneity of the vadose zone and imaging the water content distribution (within the limitation of their resolution: e.g., Daily et al. 1992, Day-Lewis et al. 2005), the quantifications of dynamic processes related to water flow and (reactive) transport still remain an

important challenge.

In vadose zone hydrogeophysics, the most promising approaches to tackle these challenges rely on using multiple methods and integrating the measured data in 4D numerical simulations with joint inversion strategies together with appropriate petrophysical knowledge (e.g., Hubbard and Linde, 2011, table 1). Surface-based and cross-borehole imaging of the water



content in the vadose zone through measurements of the electrical conductivity or dielectric permittivity distribution is well
established (e.g., Carrière et al., 2015, 2022, table 1.B.5). The electrical conductivity can be obtained using low-frequency
electrical and electromagnetic methods such as Electrical Resistivity Tomography and Induced Polarization (e.g., Kemna et
al., 2012; Revil et al., 2012). Higher frequency methods such as the Time Domain or Frequency Domain Electromagnetics can
also be employed from the surface (e.g., Pelerin, 2002) or the air (Auken et al., 2020) allowing to cover large areas in a limited
amount of time, but show generally a poor vertical resolution at the scale of the vadose zone. Electrical and electromagnetic
methods are well established in static conditions but also in monitoring applications (e.g., Singha et al. 2015). The main
limitation of these methods is the downside of their integrative nature: a limited resolution that masks heterogeneities and can
mislead quantitative estimation of water content or solute concentration using petrophysical relationships established in the
laboratory (Day-Lewis et al., 2005; Jougnot et al., 2018) as already discussed in section 2.2. Combining several methods (e.g.,
Blazevic et al., 2020, table 1.B.1) and improving the petrophysical-based approaches  (e.g., Day-Lewis et al., 2017) is needed
to move towards a more quantitative use of electrical and electromagnetic methods.

Ground-Penetrating Radar (GPR) is the most developed geophysical method to obtain the water content at high spatio-temporal
resolution through the dielectric permittivity (e.g., Huisman et al., 2003, Roth et al. 1990; Klotzsche et al., 2018; Looms et al.,
2008). Time-lapse studies using GPR to monitor water infiltration can provide insights about the hydrodynamic (e.g., Léger
et al., 2014; Klotzsche et al., 2019a) and transport properties (e.g., Haarder et al. 2015) of the vadose zone. Commonly,
geophysical studies of vadose zone processes use 1D approximations although the 3D nature of the heterogeneous vadose zone
can obviously result in lateral flow (Scholer et al., 2012).

Nuclear Magnetic Resonance (NMR) allows to obtain a signal that is directly related to the quantity of water in the subsurface.
It is based on the resonance of the magnetic moment of the protons from water molecules. NMR can be used from the surface
or boreholes to infer the water content in the vadose zone (e.g., Schmidt and Rempe, 2020). Recent works have also shown its
value for monitoring the water dynamics through time-lapse measurements (e.g., Mazzili et al. 2020, Lesparre et al. 2020).

From established P-wave velocity tomography (e.g., Bradford, 2002) to more recent imaging of surface wave velocities and
Poisson ratios (e.g., Pasquet et al., 2015, table 1.B.4), active seismic methods are developing toward a much more quantitative
characterization of the water content distribution (e.g., Pride, 2005). Surface waves appear promising for monitoring of water
content dynamics in the vadose zone (Dangeard et al., 2018, table 1.B.4) and combining time-lapse imaging of seismic
tomography and ERT will allow providing a more quantitative imaging (Blazevic et al., 2020, table 1.B.1). Figure a show the
acquisition set-up used by Blazevic et al. (2020) to jointly monitor the infiltration of water in a delimited area above a pit
instrumented with TDR (Fig. Figure b). Figure c and Figure d show the developments of the relative changes in ERT and
seismic inversions results during the infiltration along the North-South line, respectively. One can see the preferential water
flow from North to South, indicating lateral flow in the vadose zone. This use of complementary methods, in terms of resolution
and sensitivity to properties (electrical conductivity and mechanical properties), opens up new perspective such as joint
inversion (e.g., Doetsch et al., 2010) and petrophysical-based inversion (e.g., Wagner et al., 2019)



Passive seismic is also receiving increasing attention as ambient noise can be used as a source to monitor hydrosystems. Recent works on seismic noise monitoring have been conducted using ballistic waves to monitor water table variations (Garambois et al., 2019). The development of distributed acoustic sensing will allow the acquisition of denser and larger-scale monitoring data in this direction (e.g., Zhan, 2020).

Another passive method that is increasingly used in the vadose zone is the self-potential method (e.g., Revil and Jardani, 2013). It consists in measuring naturally occurring electrical voltages that results from various coupling mechanisms, for instance, electrokinetic coupling when water flows in a porous medium or in a fractured system (e.g., Jougnot et al. 2020; Robert et al, 2011). A promising approach to monitor water movement in the vadose zone is to implant the SP electrodes in the ground at different locations and depths, the measured signal is then integrated over the volume delimited by the electrodes, allowing vertical and lateral monitoring. Recent works of SP monitoring have shown the usefulness of SP to monitor infiltration (Jougnot et al. 2015, Hu et al., 2020, see table 1.C.6) and root water uptake (Voytek et al., 2019). These works have shown the need for further improving petrophysical models to shift the use of SP towards a more quantitative paradigm.

Lastly, gravity is a well-established passive geophysical method that is suitable to monitor water movement in the vadose zone (Fores et al., 2017, 2018, table 1.C.4). Due to its very large footprint that integrate the density distribution from the center of the Earth, there is a crucial need for more accurate and sensitive gravimeters, such as e.g. quantum absolute gravimeters (Cooke et al., 2021). The non-uniqueness of gravity signals requires the inclusion of complementary information (e.g. geodetic or hydrological data) for signal separation. Time-lapse gravimetry has been used to identify and constrain subsurface water storage changes, e.g. in artificial recharge facilities (Kennedy et al., 2016), to separate precipitation and groundwater mass signals (Delobbe et al., 2019), to locate karst storage dynamics (Pivetta et al., 2021) and identify evapotranspiration patterns (Carrière et al., 2021). Further data acquisition procedures and treatments that enhance sensitivity to local processes (e.g., gravity gradients and hydrological modelling coupled with gravity measurements (Cooke et al., 2020, table 1.C.5) are needed to provide more quantitative interpretations.

In summary, the 3D nature of the subsurface is made even more complex in the vadose zone by its partial saturation and the evolution of the water phase, both in term of saturation and spatial distribution, as well as by the presence of root systems and microorganisms. While quantitative assessment of aquifer recharge can often be achieved by integrative approaches, the spatial and temporal distribution of the water content in the vadose zone is crucial for assessing water available to the plants through their roots, to identify preferential flow paths for contaminants or infiltration in karstic aquifers, or for optimizing irrigation practices.

## 2.4 Groundwater – surface water interactions

The interface between groundwater (GW) and surface water (SW) is a structurally complex, dynamic transition zone that modulates fluxes of water, solutes and heat between the two adjoining compartments (Lewandowski et al., 2019). These fluxes in turn affect several processes that are relevant for the management of water quantity (e.g. water supply via bank filtration,





groundwater recharge), quality (e.g. pollutant attenuation, nutrient transformations – eutrophication) and aquatic ecology (e.g. environmental flows, habitat and refugia) (Fig. 5).

Exchange and turnover patterns in the GW-SW transition zone are defined by nested spatial controls ranging from regional topography and geology (Winter 1999) to local variability of streambed permeability (Kalbus et al. 2009, Irvine et al. 2012; Tang et al., 2017) and morphology evolution (Trauth et al. 2015; Partington et al., 2017), the spatial arrangement of subsurface hydrofacies (Fleckenstein et al. 2006, Frei et al. 2009; Carlier et al., 2018) and their anisotropy (Gianni et a., 2018) and reactive zones (Frei et al. 2012, Loschko et al. 2016). Temporal dynamics are mainly imposed by the surface water system (Dudley– Southern and Binley 2015, Trauth and Fleckenstein 2017, Song et al. 2020), as SW heads can change significantly over short – event – time scales while head changes in GW occur more gradually (e.g. at seasonal time scales). The resulting fluctuations in hydraulic gradients affect subsurface mixing (Hester et al. 2017, Bandopadhyay et al. 2018), as well as transit times and reactive turnover (Zarnetske et al. 2012, Trauth and Fleckenstein 2017). Understanding the linkages and feedbacks between spatial and temporal controls of flow and turnover is needed for a sound management of coupled GW-SW systems and their ecosystem services (Hester and Gooseff 2010, Morén et al. 2017, Hester et al. 2018). This understanding clearly hinges on innovative methods and the ability to test them at dedicated field sites. The insights gained at these sites may provide a basis for simplifications and generalizations, which can be used to improve modeling concepts for management of GW-SW systems beyond the specific field sites.

Recent years have seen significant advances in methods and technologies for the characterization and simulation of coupled GW-SW systems. Methods particularly suited for the study of GW-SW interactions, to name a few, include in situ and high resolution sensing of temperatures (Constantz 2008, Vogt et al. 2010) and solute concentrations (Blaen et al. 2016, Brandt et al. 2017), tracer techniques to characterize exchange flows (Mallard et al. 2014; Schilling et al., 2017a; Popp et al., 2021), transit times and reactions (Schmidt et al. 2012, Knapp and Cirpka 2017), as well as process-based, integrated modeling of coupled GW-SW systems (Schilling et al. 2017b, Trauth and Fleckenstein 2017, Broecker et al. 2019) and geophysics (McGarr et al., 2021). Here, we briefly discuss some key research fields related to the heterogeneous and dynamical nature of the GW-SW interaction, which have, and likely will continue to contribute to an improved understanding of GW-SW interactions.

The use of heat as a natural tracer has become a popular tool to characterize GW-SW exchange patterns due to the natural temperature differences between GW and SW and the relative ease and accuracy of temperature measurements using standard sensors. This field has evolved significantly since some of the earlier seminal works (Stonestrom and Constantz 2003, Schmidt et al. 2006) and has embraced novel technologies such as DTS (Krause et al. 2012, Rose et al. 2013) and hand-held (Glaser et al. 2016, Marruedo Arricibita et al. 2018) or airborne infrared imagery (Lewandowski et al. 2013). The suite of methods available today allows for high-resolution assessment of temperatures in space and time for a qualitative mapping of GW-SW exchange patterns (Anibas et al. 2011, Krause et al. 2012) or a quantification of exchange fluxes (Schornberg et al. 2010, Munz and Schmidt 2017). Temperature data can further be used to constrain and improve numerical models of coupled GW-SW systems (Munz et al. 2017). Due to their relative ease of use, heat-based methods have become a robust and standard tool to characterize GW-SW exchange patterns. New opportunities may arise from a smart combination of different techniques



(Tirado-Conde et al. 2019) or the use of actively heated fiber optics (Simon et al. 2021), a technique (see also section 2.1) that has been used punctually to quantify streambed flow dynamics in zones of groundwater upwelling (Briggs et al. 2016), and

which is in current development for quantifying GW-SW exchange patterns along stream sections (Simon et al., 2022).

The concentration of oxygen is a key variable that defines the redox state of the transition zone between GW and SW and with that the potential for important reactions like denitrification (Zarnetske et al. 2012). Understanding the dynamics of oxygen consumption is therefore important for evaluating nutrient turnover in hyporheic and riparian zones (Marzadri et al. 2012, Trauth et al. 2015, 2018). A key reaction consuming oxygen in these zones is aerobic respiration, which has been shown to

depend on transit times (Zarnetzke et al. 2011a, Diem et al. 2014) and the availability of labile organic carbon as the main electron donor (Zarnetske et al. 2011b). Field deployable optode-based oxygen sensors have enabled high-resolution measurements of oxygen concentrations in time (Diem et al. 2014, Vieweg et al. 2016) and space (Brandt et al. 2017) allowing for robust assessments of respiration dynamics at the GW-SW interface (Vieweg et al. 2016). Based on such data, the strong temperature dependence of aerobic respiration rates has been demonstrated (Diem et al. 2014), which may dominate turnover

rates compared to the effects of variable transit times (Nogueira et al.,2021a). Similar effects were found to affect complex spatio-temporal patterns of riparian denitrification, which seem to be jointly controlled by hydraulically driven variability in exchange fluxes and transit times, supply of organic carbon as an electron donor from stream water and riparian sediments and seasonal temperature variations (Trauth et al. 2018, Lutz et al. 2020, Nogueira et al. 2021b). Besides high-resolution data sets of key variables like oxygen concentration, it is often the combination of these rich data sets with innovative methods for

analysis and modeling (e.g. Diem et al. 2014, Lutz et al. 2020, Nogueira et al. 2021a, 2021b) that advances our mechanistic understanding of the processes and feedbacks that define the functionality of GW-SW interfaces.

Another promising and still evolving field in the area of GW-SW interactions has been the use of mechanistic models in explorative mode to test hypotheses and to investigate process interactions and feedbacks (Fleckenstein et al. 2010, Brunner et al. 2017). Important insights into physics of flow, transport and turnover processes in the hyporheic zone (HZ) have been

gained based on such modeling studies. This includes the effects of ambient groundwater flow on hyporheic exchange (Cardenas and Wilson 2007, Trauth et al. 2013, 2015), intermeander and parafluvial flows (Boano et al. 2006), reactions and turnover in the HZ (Boano et al. 2010, Trauth et al. 2014, 2015), effects of geologic heterogeneity on hyporheic flows and reactions (Laube et al. 2018, Bardini et al 2013) as well as the influence of stream flow dynamics on hyporheic exchange and turnover (Trauth and Fleckenstein 2017, Singh et al. 2020). Similar modeling studies have been conducted for riparian zones

and river corridors addressing aspects such as the effects of streamflow variations on riparian solute turnover (Gu et al. 2012), effects of bank filtration processes on solute mobilization from riparian zones (Mahmood et al. 2019),the effects of riverbed heterogeneity on GW-SW exchange patterns (Tang et al. 2017) or the presence and dynamics of unsaturated conditions at the stream-aquifer interface (Schilling et al. 2017). Some studies have also used modeling experiments to address mixing processes at GW-SW interfaces, which are important for mixing-dependent reactions (Hester et al. 2017). These studies have addressed

effects of flow geometries and hydraulics on mixing in hyporheic zones (Bandopadhyay et al. 2018; Lee et al., 2021) or effects of geologic heterogeneity at the groundwater-seawater interface on calcite dissolution and karstification (De Vriendt et al.



2020) as discussed in more detail in section 2.2. Advances in modeling capabilities, including a seamless, integral simulation of GW-SW systems (Broeker et al. 2019; Li et al., 2020), together with rich data sets from highly instrumented field sites to confront these models with, will help to improve our mechanistic understanding of GW-SW interactions. Such approach will

provide a solid basis for generalizing and developing new concepts allowing the use of more parsimonious models, focusing on the dominant processes, which could be more largely applied for management-relevant scales (Gomez-Velez and Harvey, 2014).

In summary, the functionality of the GW-SW interface is clearly defined by processes operating in 4D. A suite of field methods and modeling tools exists to characterize process patterns and dynamics. New methodological developments as well as

innovative improvements to existing methods, and their systematic use at dedicated field sites, will open up new opportunities to advance our mechanistic understanding of GW-SW interactions and to develop more parsimonious models, which are needed to develop appropriate measures for the management of coupled GW-SW systems.

## 3 Numerical methods development for 4D data integration and inversion

Numerical representation methods and numerical techniques have become essential tools in both understanding and forecasting

subsurface models (e.g., Karatzas, 2017). Most common software suites in hydrogeology allow to model the subsurface using properties distributed in 3D. The temporal derivatives being an essential component of underlying physical equations, the transient character of hydrogeological processes is most often already included. When a mathematical formulation of the process exists, numerical methods allow simulating the response to any scenarios as long as the distribution of the involved hydrogeological parameters is provided. Specific models can handle non-linearity related to time-varying properties as

illustrated by coupled models for unsaturated flow or geomechanics (Simunek et al., 2018; Davy et al., 2018). However, next to accuracy related to solvers and their parameterization (numerical dispersion, instability, non-convergence), numerical models remain dependent on the accuracy of the underlying mathematical representation of the modelled process. Previous sections have highlighted that experimental work remains necessary to characterize complex processes such as mixing and transport or GW-SW interactions (Heyman et al., 2020).

Nevertheless, several challenges remain to properly simulate heterogeneous groundwater reservoirs and their dynamics with numerical models. As highlighted in the previous sections, a key aspect is to feed the models with the appropriate data input (e.g., Schilling et al., 2019). Subsurface processes are influenced by the heterogeneity in subsurface properties. If the latter is essential for the purpose of the model, this should be reflected in the data input (e.g., Guillaume et al., 2019). Even with powerful computers, simulating the transient response of a catchment-scale model with limited resolution might still take

several hours, days or even weeks (e.g., Hayley et al., 2014). The same is true for high-resolution geophysical models such as full-waveform GPR or 3D electromagnetics (e.g., Oldenburg et al., 2013; Klotzsche et al., 2019b; Zhou et al., 2020; Haruzi et al., 2022) and coupled approaches (Coulon et al., 2021). Including geophysical data in hydrogeological models is thus even more challenging. Surrogate models can be used to speed up simulation processes, but their accuracy remains dependent on





the training process, which can be problematic in highly heterogeneous media (Linde et al. 2017; Köpke et al., 2018; Mo et
al., 2020). Small-scale heterogeneity is present in many geological contexts (e.g., Bayer et al., 2011). Even if it is relevant for
the objective of the model, it must thus often be neglected due to limited computational resources. This sometimes leads to
unrealistic outputs, in particular for transport processes (e.g., Hoffmann et al., 2019). This may be addressed by upscaling
heterogeneous systems and defining equivalent properties at larger scale, but this is very challenging for transport processes
with multiple physical scales and non-equilibrium phenomena (Li et al., 2017; Icardi et al., 2019).

Even if simulating small-scale heterogeneity would become possible, the actual distribution of properties is always unknown.
The limited amount of noise-contaminated data does not allow to unequivocally recover this distribution through inverse
modeling (Zhou et al., 2014), even when distributed geophysical data is available (Hermans et al., 2015a; Mari et al., 2009,
table 1.D.1). On the one hand, deterministic approaches have to simplify the parameter estimation problem to make it well-
posed (e.g., smoothing, zonation or pilot-points) and are therefore, limited, for tackling uncertainty related to 4D processes.
On the other hand, stochastic approaches such as Markov chain Monte Carlo (McMC) methods (e.g., Vrugt et al., 2013), using
more or less complex and realistic geostatistical representations of the heterogeneity and more or less wide prior distribution
(Linde et al., 2015b), often require thousands to millions of simulations to converge, especially when spatial uncertainty is
included (e.g., De Pasquale et al., 2019). The transient aspects are also commonly simplified, due to a lack of data or the
simplification of boundary conditions. In most applications, resolving numerically a time-dependant system of partial
derivative equations based on spatially distributed parameters in 3D with a high spatial and temporal resolution to represent a
relatively large system remains utopic. It is therefore of uttermost importance to understand which simplifying assumptions
can be applied without degrading the predictive capability of the model (Schilling et al., 2019).

Fractured aquifers are even more complicated to model. Fracture networks have complex 3D geometries and are, therefore,
difficult to characterize from mostly 1D or 2D data (Le Goc et al., 2017; Day-Lewis et al., 2017). Modeling flow in fractures
requires another scale compared to matrix flow, which brings additional challenges in terms of gridding (Schädle et al., 2019)
and inversion (Ringel et al., 2019). Combined with a higher degree of heterogeneity and uncertainty than in porous media, it
increases the above-mentioned issues preventing efficient modeling.

Nevertheless, recent advances shed light on some innovative solutions to tackle those problems. For fractured aquifers, recent
studies in fracture modeling such as realistic flow characterization and mechanical coupling using discrete fracture networks
(Davy et al., 2018; Maillot et al., 2016; Lei et al., 2017), innovative inverse methodologies (Pieraccini, 2020) and
characterization techniques (Dorn et al., 2013; Shakas and Linde, 2017; Molron et al., 2020, 2021, table 1.B.2) pushed forward
our ability to account for the complexity of fractured media.

Cloud computing combined with increasing computational power should allow to model the subsurface at a higher 4D
resolution for an increasing number of applications in the future, including for (small) consulting companies and field
practitioners (Hayley, 2017; Kurtz et al., 2017). The ongoing efforts for coupling different simulators, both in hydrogeology
and hydrogeophysics will also favor the incorporation of larger, more informative data sets in modelling efforts (e.g., Commer
et al., 2020).





Although Bayesian methods such as McMC are widely recognized in the scientific literature for inversion, prediction and uncertainty quantification (Ferré, 2020), they have not been widely adopted by practitioners because of their computational burdens, especially for complex uncertain geometries. Recent developments in machine learning such as deep neural networks (DNN) have shown that complex spatial patterns can be efficiently reduced to a manageable number of dimensions. DNNs allow to simplify complex subsurface models with millions of cells to a few tens of dimensions, while maintaining their geometrical complexity represented by a prior parameter distribution. This opens the possibility to apply McMC or global optimization methods at a reasonable cost as recently demonstrated by Laloy et al. (2018) and Lopez-Alvis et al., (2021) (see Figure ). Since the parameterization of the prior is a key to obtain realistic solutions, the identification of realistic geological scenarios through falsification techniques (e.g., Hermans et al., 2015a; Linde et al., 2015a; Lopez-Alvis et al., 2019, 2022) can further improve stochastic inversion by reducing the range of the prior.

Another recent innovation is to propose physically-based geostatistical upscaling allowing to translate the small-scale spatial uncertainty at a larger scale (Benoit et al., 2021). Combining those recent advances with accurate fast approximations of the forward model (Linde et al., 2017) should allow for more accurate representations of spatial variability within affordable computational time. As an alternative, approximations of the inverse problems using Ensemble Kalman generator (e.g., Nowak, 2009, Bobe et al., 2020, Tso et al., 2021) or normalization and linearization approaches (e.g., Holm-Jensen and Hansen, 2019) can also provide a relatively fast, though realistic, approximation of the solution to the inverse problem.

Alternatively, recent studies have proposed to investigate more prediction-oriented strategies to simulate complex hydrogeological systems (Sun and Sun, 2015; Ferré, 2017; Scheidt et al., 2018). For example, Bayesian Evidential Learning (BEL) proposes to use a set of simulations from realistic numerical models including the 4D complexity to learn a statistical relationship between a predictor (data set) and a target (model output, see Figure ), making it a model-driven machine-learning approach, circumventing the challenges of non-linear inversion while providing a proxy for global sensitivity analysis (Hermans et al., 2018). The statistical learning requires reducing the dimensionality of the problem, so that part of the complexity may remain unresolved (Park and Caers, 2020). If such a statistical relationship exists, this approach has the advantage to require only a limited amount of simulations to derive the posterior distribution of the prediction, typically only a few hundreds to a few thousands. The latter are independent and can be fully parallelized. Propagating noise in the statistical relationship is also straightforward (Hermans et al., 2016). Such direct forecasting is possible because predictions often have a much-lower dimensionality than models. Nevertheless, when the data-prediction relationship is complex and highly non-linear, BEL might overestimate uncertainty (Michel et al., 2020a), for instance when the prior uncertainty is large (Hermans et al., 2019). In such a case, classical inversion might still be needed (Scheidt et al., 2018). Recent advances have shown that BEL can also estimate the model parameter distributions and be used as a more traditional inversion technique (Yin et al., 2020; Michel et al., 2020a). However, such more advanced applications require further development of appropriate tools to identify highly non-linear relationships (Park and Caers, 2020) which will inevitably come at a larger computational cost (Michel et al., 2020b).



Although these recent developments are promising solutions to integrate large 4D data sets within efficient simulation and inversion framework to forecast the behavior of aquifers, they still need to be more widely evaluated and used, including for complex field cases. Their validation with case studies in different contexts will demonstrate if they are adapted for the incorporation of the 4D complexity of hydrogeological processes.

## 4 The added-value of instrumented field sites

Previous sections have demonstrated how the acquisition of spatially and temporally distributed data is crucial for hydrological processes understanding. Observation gaps can severely impede our understanding of subsurface processes, while the acquisition of dense 4D data sets may provide new insights into internal mechanisms and process hierarchies. These insights can enable the identification of dominant processes, which govern a specific response of the system and in turn allow for conceptual simplifications. For example, they can help us to characterize with a higher accuracy the influence of heterogeneity and to understand how the small-scale processes must be upscaled to larger scales. More importantly, an increase in the number of applications including 4D investigations can help us to identify in which hydrogeological contexts, at which scales, and for which purposes the characterization of the role of spatial heterogeneity and dynamic processes is required; or which data and at which sampling rate is necessary to properly account for the underlying subsurface complexity. This would then allow to release the need for dense 4D data for practical applications with limited budget based on sound scientific observations.

Over the last decades, highly instrumented field sites have been equipped to explore and monitor subsurface processes, often with detailed geological models available (e.g. Bogena et al. 2018). Long-term observations of subsurface environments, typically more than ten years, are motivated by the broad range of responses and residence times of these systems that provide resilience to hydrological systems to environmental changes. When characterizing processes beyond the laboratory scales, the exhaustive 4D characterization of processes becomes an insurmountable task. While it is illusory and not necessary to equip all subsurface systems with 4D imaging techniques, the use of high spatial and temporal resolution techniques in few highly instrumented field observatories during passive monitoring and active experiments may be a key step to upscale laboratory observations to a scale relevant for practical applications. In particular, these datasets should allow unraveling the importance of 4D dynamics and the influence of different processes on hydrogeophysical signals. We argue that these field observatories provide a key step between theory and laboratory experiments on one side and practical field applications on the other side. Available datasets related to such sites often combine fixed geophysical and hydrogeological sensing systems for the short-term monitoring of experimental campaigns, but also for the long-term monitoring of natural systems required to characterize physical and chemical heterogeneity and target hotspots, which cannot be easily accessed by classical observations. Related studies often provide parametrized hydrological models for the interpreration.

A non-exhaustive list of time-lapse hydrogeological and hydrogeophysical datasets available online, selected based on their on-line availability and relevance for illustrating this study, is given in Table 1. As can be seen, 4D data openly available are still scarce. Because such extensive spatially and temporally resolved imaging can only be achieved on few sites, we argue



that it is important to archive and share such datasets. In particular, these data are critical to i) test and validate model hypotheses and predictive capabilities, ii) to develop appropriate inverse modeling approaches adapted to the high level of heterogeneity of subsurface environments, iii) evaluate the added value of different imaging techniques in order to optimize the design of monitoring strategies on other sites. Table 1 will be made available through the website https://hplus.ore.fr/en/database/4d-hydrogeology-dataset, which will be open to contributions to enrich the datasets with new data.

## 5 Concluding remarks

In this paper, we have illustrated that advances in our understanding of complex subsurface processes hinge on our ability to observe/image key parameters and state variables with the relevant spatial and temporal resolution.

Although spatial heterogeneity and temporal variations by default influence all the processes occurring in the subsurface, a full 4D characterization coupled with a numerical model is not always necessary. For example, the management of water resources based on meteorological and production data using simple water balance approaches has been applied successfully in many contexts for decades. We may obtain reasonable estimates of groundwater volumes and fluxes from sparse hydrogeological data, so that a fine scale characterization of the heterogeneity might not be needed or working in steady-state conditions might be sufficient. Estimating an average recharge rate for by-passing the complex processes occurring in the vadose zone has been proved to be efficient in many contexts.

Nevertheless, key applications where the coupling of spatial heterogeneity and temporal fluctuations might be essential include heat and solute transport, mixing and reaction processes occurring in the subsurface and at the interface of rivers. Identifying spatial and temporal fluctuations of groundwater flow is, thus, key to unravel important processes such as the transport of contaminant or the biodegradation of pollutants. In specific applications, data availability is always limited, due to budget, time and space constraints, and conceptual simplifications are required. In this context, field observatories linked to open database and techniques dedicated to the acquisition of spatially and temporally resolved data sets are essential to built appropriate models based on spatially and temporally resolved measurements.

New technologies are advancing our ability to close observation gaps and image the subsurface heterogeneity and dynamics for processes of relevance in hydrogeology. The development of techniques and sensors directly sensitive to transient fluxes (finite-volume point dilution method, active-DTS), constitutes a major advance towards understanding the complex processes taking place in the subsurface, although further developments are still needed to combine these techniques with classical geophysical methods to allow full 4D imaging. Similarly, the combination of classical salt and dye tracer with heat and dissolved gasses has the potential to further discriminate the different transport processes and to understand the exchanges between mobile and immobile water, both in porous and fractured media, but also for surface-subsurface interactions. Such data collected across scales and with dense networks could lead to a better description of the processes and the development

of new mechanistic models as well as the proper definition of effective parameters and upscaling approaches to do away with
the complexity at larger scales.

Considering the spatial heterogeneity of the subsurface quickly requires the use of efficient and reliable numerical models. Including small-scale heterogeneity automatically calls for high-resolution models with refined grids leading to high if not unacceptable computation times. In addition, the uncertainty inherent to subsurface systems can only be properly dealt with by stochastic approaches that require many simulations to characterize the ensemble of possible outcomes. The current
computational power available for the scientific community, and for practitioners in the industry, is not sufficient to systematically tackle groundwater reservoirs with their full 4D complexity.

Simplifications, such as ignoring the small-scale heterogeneity or ignoring some transient processes, are always needed and can in many cases provide useful results for groundwater management. Nevertheless, the hydrogeological community is still often facing model predictive outcomes that are not consistent when validation data become available. Even though the 4D
complexity is not always the cause, it can probably explain why some models have poor predictive capability. Ideally, hydrogeological conceptual models should initially consider the 4D complexity of the system, and only deviate from this rigorous description when there is no significant effect on the prediction or decision-making process. Such conceptual simplifications of hydrogeological systems should be based on strong experimental or numerical evidences, and should not constitute the default hypothesis because of lack of data or computational power. In that sense, the existence of large 4D data
sets linked to field observatories and models can only be beneficial for the community. Similarly, the more systematic use of Monte Carlo simulations, prediction-oriented approaches or global sensitivity analyses, although computationally expensive, can provide the necessary background information, at least for processes that can be properly characterized by mathematical models.

### Acknowledgement

This work has received funding from the European Union's Horizon 2020 research and innovation program under the Marie Sklodowska-Curie grant agreement number 722028 (ENIGMA ITN). We thank all the ENIGMA beneficiaries and partners for their direct and indirect contributions to this article.

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





**Figure 1:** Illustration of the 4D nature of hydrogeological processes inaccessible from punctual data only: a) imaging subsurface fluxes and their contribution to surface fluxes through the combination of borehole and surface imaging; b) monitoring water content and temperature fluctuations for quantifying preferential flows and heat transfer; c) characterizing tracer motion to elucidate transport processes and related parameters; d) analyzing and upscaling pore-scale signals produced by microscale reactive transport processes.



A. Darcy scale helical flow in anisoptropic permeability fields (centimeter to meter scale)

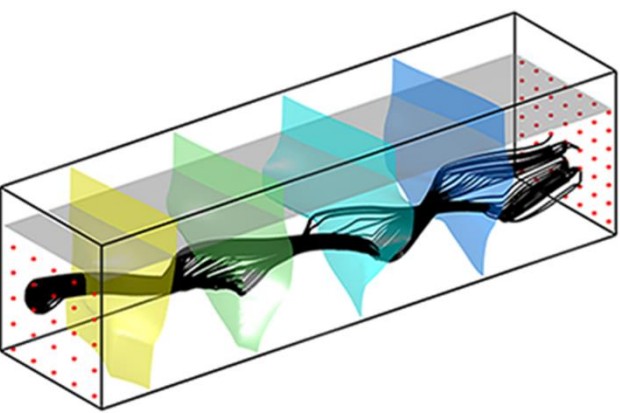

B. Chaotic flow dynamics at the pore scale in granular media (millimeter to centimeter scale)

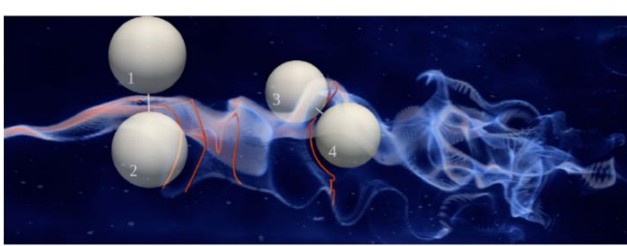

**Figure 2: A. Experimental evidence of helical flow at Darcy scale (top, modified from Ye et al., 2015): the black lines show simulated streamlines, the colored surfaces iso-pressure surfaces and the red dots are experimental sampling points. B. Experimental evidence of chaotic mixing at the pore scale in three-dimensional porous media (bottom modified from Heyman et al., 2020): the color field shows the concentration of a continuously injected fluorescent dye, the red line is the intersection of the plume with a 2D plane transverse to the mean flow, which shos the stretching and folding patterns that produce chaotic mixing, the grey spheres represent selected grains in the bead pack that create the first successive folding of the plume.**





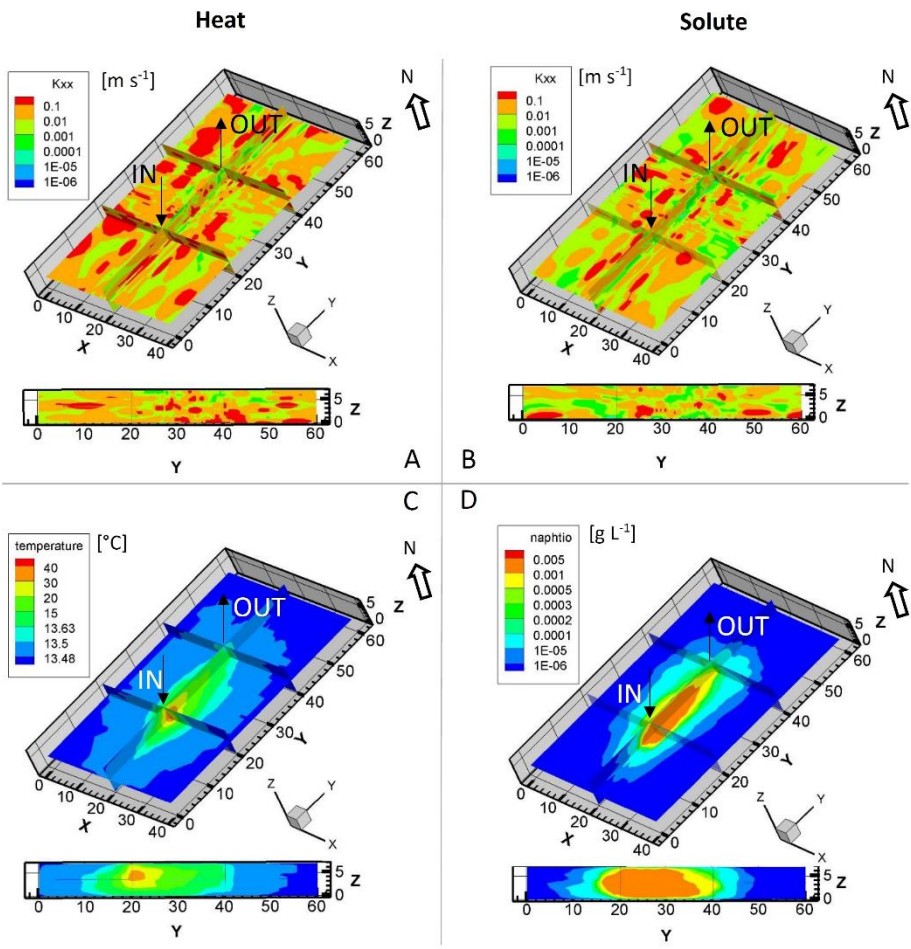


**Figure 3. Hydraulic conductivity fields (A-B) and related simulations (C-D) obtained by inversion of temperature (A-C) and solute tracer concentration data (B – D) from heat and solute tracer experiments (modified from Hoffmann et al., 2019, table 1.A.1).**





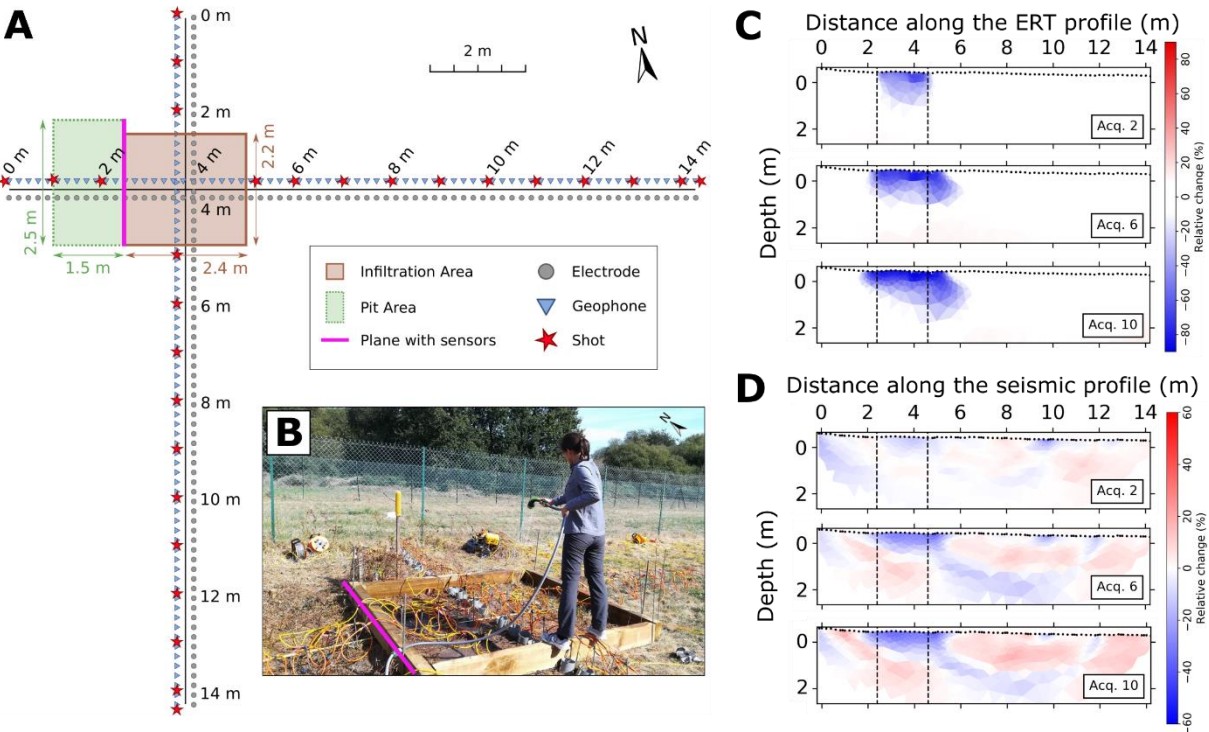

**Figure 4.** **(A) ERT and seismic lines that cross in the infiltration area. (B) Picture of the infiltration test. Temporal evolution of relative change in properties inferred from (C) the ERT and (D) the seismic data at different times (i.e., successive acquisitions), respectively (modified after Blazevic et al., 2020, table 1.B.1).**






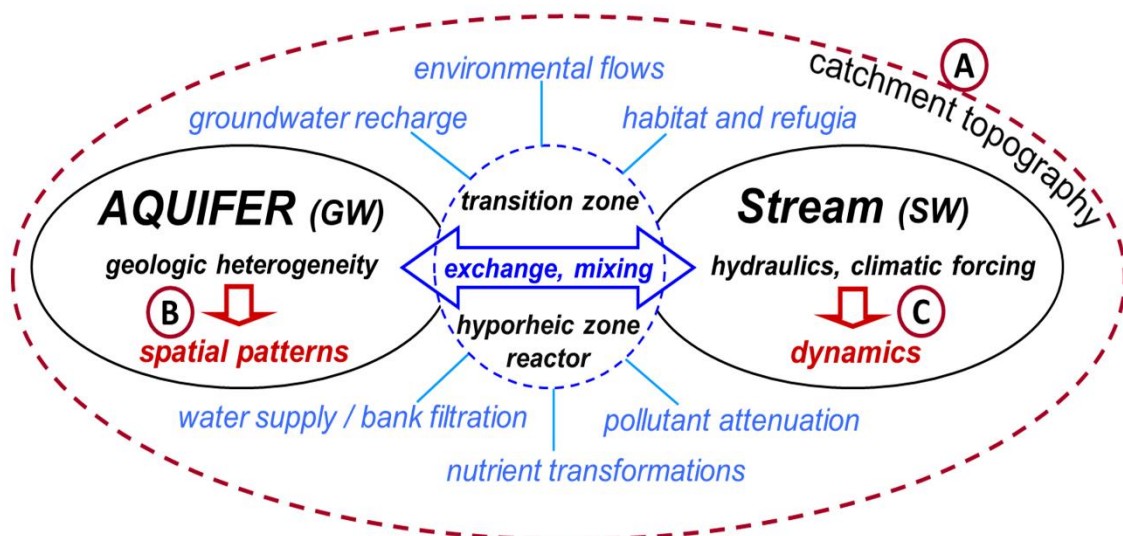

**Figure 5. Conceptual depiction of the GW-SW interface and the nested controls of spatio-temporal patterns of exchange and solute turnover in the transition zone between GW and SW (using a river-aquifer system as an example). A: primary spatial controls defined by regional topography and geology, B: secondary spatial controls defined by local aquifer and stream bed geologic heterogeneities as well as streambed morphology, C: temporal controls caused by river flow dynamics. Processes and management aspects that are affected by these exchange and turnover patterns are shown in light blue.**



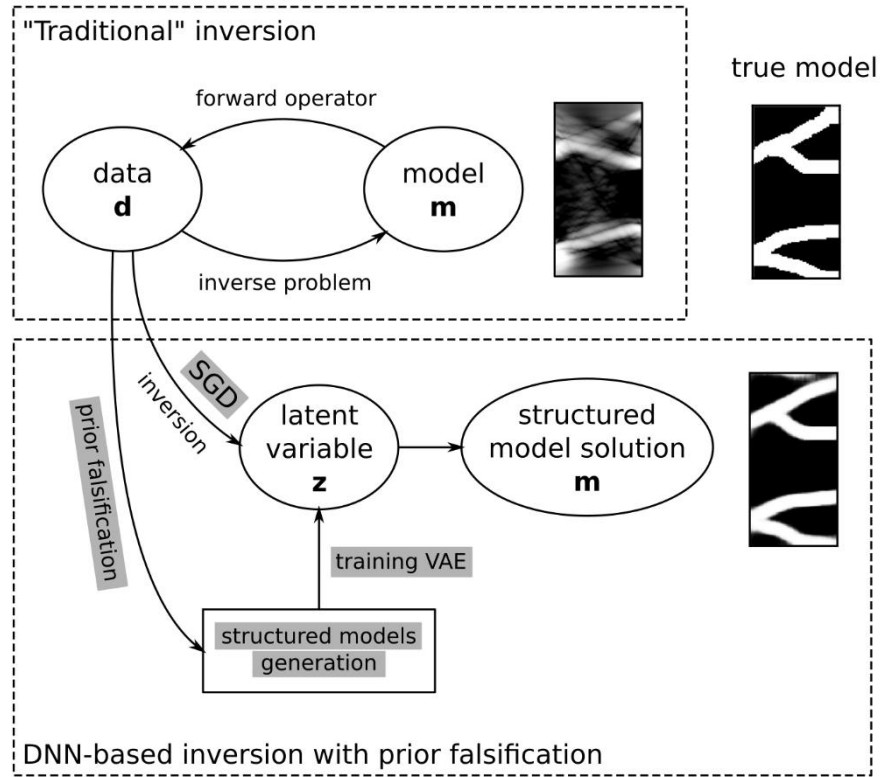

**Figure 6: Adapted deep neural network (DNN) based inversion scheme using a variational autoecnoder (VAE) to represent complex geological structure in a latent space represented by sparse variable in which a stochastic gradient-descent (SGD) can be applied to converge towards the a more geologically realistic solution modified from Lopez-Alvis et al., (2022).**





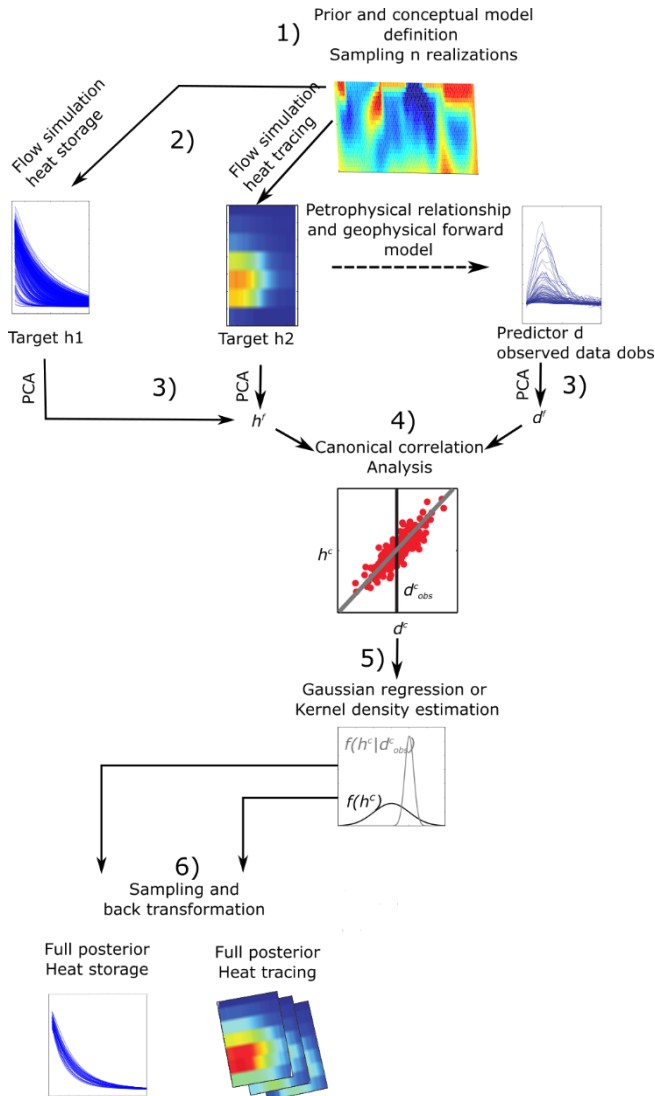

**Figure 7: Framework of Bayesian Evidential Learning applied to the prediction of heat storage in an aquifer from a tracer experiment monitored by geophysics. 1. A prior model of the spatial heterogeneity is defined and sampled. 2. Both the predictor (data set such as a tracer experiment) and the target (prediction of heat storage) are simulated. For the predictor, a petrophysical transform is used to generate the corresponding geophysical data. 3. Dimension reduction techniques are applied to both data and prediction. 4. A statistical relationship between data and prediction is learned. 5. The posterior distribution of the prediction for field observed data is generated using the statistical relationship. 6. The posterior is backtransformed in the physical space (modified from Hermans et al., 2018)**



**Table 1: Selection of datasets available on-line for multi-dimensional hydrogeological system characterization and monitoring. Datasets were selected based on their on line availability, their link with a dynamic component of the subsurface and their relevance for illustrating the paper. This table will be available on line https://hplus.ore.fr/en/database/4d-hydrogeology-dataset with the possibility to contribute to enrich the datasets with new data.**

| Dimension | Dataset | Site | Link to dataset | Publications |
|---|---|---|---|---|
| **A+ 3D + time** | **A.1 Time-lapse geophysical monitoring of heat transport**: 2D time-lapse ERT cross-sections + 3D multiple nested wells data during heat tracer test | Hermalle, Belgium | https://hplus.ore.fr/en/associated-sites/enigma/data-hermalle | Hermans et al.,(2015b, 2018) (Hoffmann et al. 2019) |
| | **A.2 Time-lapse geophysical monitoring of hyporheic zone processes**: 2D surface Electromagnetic Induction + 2D time-lapse ERT (section) | Theis site, Ohio, USA | https://doi.org/10.4211/hs.69204f1ee49c4176a8aab5f4832c7b76 | McGarr et al., (2021) |
| **B.  2D + time** | **B.1 Time-lapse geophysical monitoring of water infiltration in the vadose zone**: 2D time-lapse ERT and seismic cross-sections + TDR monitoring of water content during irrigation | Ploemeur, France | http://hplus.ore.fr/en/blazevic-et-al-2020-water-data[1] | Blazevic et al. (2020) |
| | **B.2 GPR imaging of fracture opening during hydraulic test**: borehole and surface GPR+ optical televiewer + core logging during high pressure injection test | Aspö hard rock laboratory, Sweden | http://hplus.ore.fr/en/molron-et-al-2021-eg-data[1]<br>http://hplus.ore.fr/en/molron-et-al-2020-eg-data[1] | Molron et al. (2020, 2021) |
| | **B.3 GPR imaging of  tracer transport in fractured media**: borehole GPR + conductivity | Ploemeur, France | http://hplus.ore.fr/en/shakas-et-al-2017-grl-data[1] | Shakas et al. (2017, 2016) |

---

[1] with login and password provided upon request



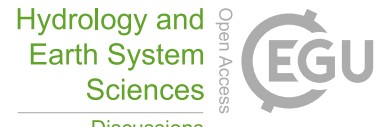

| | | | | |
|---|---|---|---|---|
| | and fluorescence monitoring during tracer test | | | |
| | **B.4 Use of $V_p/V_s$ ratio to monitor subsurface water content**: 2D time-lapse seismic cross-sections P- and surface-wave survey, SH-wave refraction acquisition, travel-time tomography | Ploemeur, France | http://hplus.ore.fr/en/pasquet-et-al-2015-nsg-data | Pasquet et al. (2015) Dangeard et al. (2018) |
| | **B.5 Time-lapse geophysical monitoring of water infiltration in karstic environment**: 2D time-lapse ERT cross-sections during rainfall events | LSBB, France | https://hplus.ore.fr/en/carriere-et-al-2022-dib-data http://hplus.ore.fr/public/requests/lsbb/LSBB_ERT_data_Carriere_24102011.csv.tgz | Carrière et al. (2015, 2022) |
| | **B.6 Time-lapse geophysical monitoring of seawater intrusion**: 2D time-lapse cross-hole ERT in coastal aquifer | Argentona, Spain | http://hplus.ore.fr/en/palacios-et-al-2020-hess-data[1] | Palacios et al. (2020) |
| **C. 1D + time** | **C.1 Nuclear Magnetic resonance monitoring of water in a headwater catchments:** Time-lapse magnetic resonance sounding | Strengbach, France | http://hplus.ore.fr/en/lesparre-et-al-2020-joh-data | Lesparre et al. (2020) |
| | **C.2 Fiber optic DTS monitoring for estimating thermal conductivity and groundwater flux in porous media:** fiber optic DTS and heat tracer experiments in a sand tank | Poitiers, France | http://hplus.ore.fr/en/simon-et-al-2020-wrr-data | Simon et al. (2020, 2022) |



| | **C.3 Fiber-optic monitoring of heat transfer in fractured media:** Fiber optic-DTS monitoring during thermal and solute tracer tests, | Ploemeur, France | http://hplus.ore.fr/en/delabernardie-et-al-2018-wrr-data[1] | De La Bernardie et al. (2018) |
|---|---|---|---|---|
| | **C.4 Use of gravimeter time series for hydrological model calibration in a karst aquifer:** Ten years gravimetry time series using iGrav superconducting gravimeter | Larzac, France | http://hplus.ore.fr\documents\requests\larzac\Larzac_gravimetry_2011_igrav002.csv.tgz<br>http://hplus.ore.fr\documents\requests\larzac\Larzac_gravimetry_2012_igrav002.csv.tgz<br>http://hplus.ore.fr\documents\requests\larzac\Larzac_gravimetry_2013_igrav002.csv.tgz<br>http://hplus.ore.fr\documents\requests\larzac\Larzac_gravimetry_2014_igrav002.csv.tgz<br>http://hplus.ore.fr\documents\requests\larzac\Larzac_gravimetry_2015_igrav002.csv.tgz<br>http://hplus.ore.fr\documents\requests\larzac\Larzac_gravimetry_2016_igrav002.csv.tgz<br>http://hplus.ore.fr\documents\requests\larzac\Larzac_gravimetry_2017_igrav002.csv.tgz<br>http://hplus.ore.fr\documents\requests\larzac\Larzac_gravimetry_2018_igrav002.csv.tgz<br>http://hplus.ore.fr\documents\requests\larzac\Larzac_gravimetry_2019_igrav002.csv.tgz<br>http://hplus.ore.fr\documents\requests\larzac\Larzac_gravimetry_2020_igrav002.csv.tgz<br>https://hplus.ore.fr/documents/requests/larzac/Larzac_gravimetry_2021_igrav002.csv.tgz | Fores et al. (2018, 2017) |
| | **C.5 Time-lapse absolute quantum gravity measurements to monitor water storage in kartic environments:** gravimeter time series | Larzac, France | https://zenodo.org/record/4279110#.YdlFqWCZOM8 | (Cooke et al, 2021) |



| | | | | |
|---|---|---|---|---|
| | **C.6 Self-potential monitoring of natural rainfall and saline tracer infiltrations at the agricultural test site of Voulund, Denmark (HOBE network):** Self-potential, time series | Voulund, Denmark | https://data.mendeley.com/datasets/6r8898657w/1 | (Hu et al., 2020) |
| **D. High resolution imaging of heterogeneity combined with tracer tests** | **D.1 3D Seismic imaging of a karstic aquifer combined with multiple cross borehole tracer tests:** 3D seismic bloc and tracer tests | Poitiers, France | http://hplus.ore.fr/en/mari-et-al-2009-ogst-data[1] <br> https://hplus.ore.fr/en/poitiers/data-poitiers[1] | Mari et al. (2009) |
| | **D.2 GPR Imaging of sand layered aquifer with multiple 2D profiles and tracer tests:** Crosshole GPR and tracer tests | Krauthausen, Germany | https://teodoor.icg.kfa-juelich.de/geonetwork/aaps/search/?uuid=ad404c9f-419a-4b14-b6e0-6ee9acd8f80e[1] | Gueting et al. (2015, 2017) |
| **E. Multiple tracer experiments** | **E.1 solute and heat tracer tests for characterizing heat transfer in fractured granite:** Convergent and push-pull tests with injection of hot water, cold water and salt | Choutuppal, India | http://hplus.ore.fr/en/hoffmann-et-al-2021-groundwater-data | Hoffmann et al. (2021b) |
| | **E.2 solute and dissolved gaz tracer tests for characterizing transport in Chalk:** Convergent and push-pull tests with heat, helium, argon, xenon and uranine | Mons, Belgium | https://hplus.ore.fr/en/hoffmann-et-al-2020-hydrogeology-of-the-chalk-data[1] <br> http://hplus.ore.fr/en/hoffmann-et-al-2020-grl-data[1] | Hoffmann et al. (2020, 2021a) |
