# Peer review of "Advancing measurements and representations of subsurface heterogeneity and dynamic processes: towards 4D hydrogeology"

_Hydrology and Earth System Sciences, 2022_

## Author Response (AR1)

Dear Editor,

Please find the revised version of our review paper "Advancing measurements and representations of subsurface heterogeneity and dynamic processes: towards 4D hydrogeology". We carefully took into account the remarks of the reviewers, and modified the manuscript accordingly.

While Reviewer 1 was very positive, the main recommendation of Reviewer 2 was to strengthen the discussion on which processes actually require high-resolution observations in time and space. To this end, we have introduced three different categories of processes and applications according to their need and potential for high-resolution space-time monitoring. We have referred to these categories at the end of each subsection and added a dedicated section discussing this question with the help of a new table and figure. We are convinced that these modifications strengthen the paper and we are thankful to the reviewers for their constructive comments.

We also draw your attention to the addition of four authors in the co-author list, G. Nogueira, A. Fernandez Visentini, J. Tirado-Conte and M. Looms. They all took part in the European network ITN-ENIGMA, which led to the writing of this paper. They contributed significantly to the intial brainstorming phase and it was the result of miscommunication if they were not included in the co-author list in the initial submission. They also contributed greatly to the revision of the manuscript. We value the contribution of these co-authors, which is equivalent to the one from the other co-authors, as described in the author contribution statement. We hope that you will accept this modification.

Below we detail our answers (in blue) to the reviewers' comments (in black).

Sincerely yours,

For the authors,

Thomas Hermans

Reviewer 1

This is a very useful 'flag in the sand' to indicate the current state of geophysical methods and to put them in context with hydrologic challenges for which they may be relevant. The authors' list is a veritable who's who of hydrogeophysics. Impressive. The only minor addition that I would have liked to see is some mention of the potential role of geophysics in the growing applications of machine learning in hydrogeology. This seems a natural fit that may well alleviate some of the problems of both fields - hydrology and geophysics. The use of ML in hydrology is limited by a lack of data that can be collected by direct means - hydrogeophysics can help to address that. Geophysics is limited because we apply very limited, simplistic petrophysical models and geophysical forward models. Perhaps a less-model-dependent interpretation with ML could alleviate that. I'll leave it to the authors to decide whether they want to open this door. But, it seems to me that a paper that is marking what the field sees as the near future might want to offer an opinion on this!

Nice work!

We thank the Reviewer for his positive assessment. The suggestion is highly relevant, and machine learning is indeed a promising way to integrate geophysical data in hydrological studies. In the original manuscript, we had discussed machine learning approaches for prediction purposes and spatial heterogeneity representation. In the revised manuscript, we have discussed recent papers on the use of machine learning for petrophysical relationship (e.g. Gottschalk and Knight, 2022) for airborne data, which opens the possibility to inform the spatial variability of aquifer at the catchment scale (or even beyond), which seems to us the most promising avenue.

See lines 617-621 in the marked manuscript.

Reviewer 2

I was quite excited when I started to read Hermans et al especially with their stated objective "to identify and discuss when, why, and for which processes and applications the characterization of dynamic hydrogeological processes is crucial."

The authors provide a massive amount of detail on groundwater fluxes, transport, mixing and reactions processes, soil moisture dynamics in the vadose zone and surface-subsurface water interactions which was interesting and useful but very little synthesis of simplified messages. I think a third to a half of the details could be striped away without losing much.  They promise to provide opinions on where and when these groundwater processes can reasonably be simplified or not but fail to deliver on this. I suggest significantly expanding this with a table that suggests criteria for when, why and what scale each of these processes can be simplified or not.

We have followed the Referee's advice to clarify our message and avoid our conclusions to be diluted in the paper. We therefore have made the following changes to the manuscript:

1. Starting from the introduction, we identified three categories of applications according to their need for high resolution space-time data.
2. In each subsection, we have added a short concluding paragraph discussing how different processes relate to these categories
3. Finally, we have added a section, a table and a figure synthesizing when, why and what scale each of these processes requires high resolution space-time imaging

The main changes can be found in lines 130-148, 227-244, 349-372, 454-464, 550-561, 657-662, 663-720 of the marked manuscript.

We have revised the text to reduce its length and avoid redundancies as well as the details related to some specific papers (we focus on general statements). Examples of such reduction can be found in lines 96-111, 187-192, 204-207, 217-218, 278-286, 295-297, 349-359, 409-411, 421-426, 480-488, 547-549 of the marked manuscript.

Instead of such useful synthesis in the modeling section they just suggest more computing power in this text:

Cloud computing combined with increasing computational power should allow to model the subsurface at a higher 4D resolution for an increasing number of applications in the future, including for (small) consulting companies and field practitioners (Hayley, 2017; Kurtz et al., 2017).

This is one of the statement in the modelling section. In the revised manuscript, we have made it clear that this is only one avenue to expand modelling to 4D, among other listed in the text. There is a synthesis of the modelling section in the last paragraph, as well as in the concluding remarks.

See lines 590-592 of the marked manuscript.

Another overall observation and critique is that scale is of outmost importance but not clearly and consistently defined and discussed in this manuscript. I think every element of the mansucript should clarify how scale plays in and I would suggest putting the processes and methods on a time-space graph like this: https://www.researchgate.net/figure/Spatial-and-temporal-scales-of-measurement-black-and-modeling-red-methods-GCM_fig1_255586996

Following the Referee's suggestion, we have introduced three categories of processes at different scales in the introduction, and linked our intermediate conclusion as well as our new section to each of these scales. They were also summarized in a table and a figure as suggested (see Table 1 and Figure 8 of the new version)..

In sum I think this manuscript would benefit from significant re-think and re-write as a major revision.

We thank the Referee for this constructive review that have helped us to improve our manuscript.

---

## Author Response (AR2)

Dear Editor,

Please find the final version of our review paper "Advancing measurements and representations of subsurface heterogeneity and dynamic processes: towards 4D hydrogeology". Compared to the accepted version, there is no variations in the content, but I made a final check for typos, and I updated the reference list to meet the requirements.

As the comment is still visible on the submission page, I also draw your attention to the addition of four authors in the co-author list (it was already the case for the revised version): G. Nogueira, A. Fernandez Visentini, J. Tirado-Conte and M. Looms. They all took part in the European network ITN-ENIGMA, which led to the writing of this paper. They contributed significantly to the intial brainstorming phase and it was the result of miscommunication if they were not included in the co-author list in the initial submission. They also contributed greatly to the revision of the manuscript. We value the contribution of these co-authors, which is equivalent to the one from the other co-authors, as described in the author contribution statement.

Sincerely yours,

For the authors,

Thomas Hermans